# BETag: Behavior-enhanced Item Tagging with Finetuned Large Language Models

## Abstract

Tags play a critical role in enhancing product discoverability, optimizing search results, and enriching recommendation systems on e-commerce platforms. Despite the recent advancements in large language models (LLMs), which have shown proficiency in processing and understanding textual information, their application in tag generation remains an under-explored yet complex challenge. To this end, we introduce a novel method for automatic product tagging using LLMs to create behavior-enhanced tags (BETags). Specifically, our approach begins by generating base tags using an LLM. These base tags are then refined into BETags by incorporating user behavior data. This method aligns the tags with users' actual browsing and purchasing behavior, enhancing the accuracy and relevance of tags to user preferences. By personalizing the base tags with user behavior data, BETags are able to capture deeper behavioral insights, which is essential for understanding nuanced user interests and preferences in e-commerce environments. Moreover, since BETags are generated offline, they do not impose real-time computational overhead and can be seamlessly integrated into downstream tasks commonly associated with recommendation systems and search optimization. Our evaluation of BETag across three datasets— Amazon (Scientific), MovieLens-1M, and FreshFood—shows that our approach significantly outperforms both human-annotated tags and other automated methods. These results highlight BETag as a scalable and efficient solution for personalized automated tagging, advancing e-commerce platforms by creating more tailored and engaging user experiences.

## CCS Concepts

• **Information systems** → **Business intelligence**; **Social tagging systems**; **Language models**; *Recommender systems.*

## Keywords

Tagging System, Large Language Models, User Behavior Modeling, Recommendation, Personalization, Information Retrieval

**ACM Reference Format:**
Anonymous Author(s). 2018. BETag: Behavior-enhanced Item Tagging with Finetuned Large Language Models. In *Proceedings of Make sure to enter the correct conference title from your rights confirmation emai (Conference acronym 'XX).* ACM, New York, NY, USA, 12 pages. https://doi.org/XXXXXXX.XXXXXXX

## 1 Introduction

In the fast-evolving world of e-commerce, businesses strive to create seamless and efficient online shopping experiences. To achieve this, vast amounts of information are collected and processed to improve customer interactions, optimize product management, and enhance operational workflows. A critical part of this process is the ability to organize and categorize products effectively, which enables customers to navigate extensive inventories with ease.

One such solution is product tagging, a crucial and common element in the current e-commerce ecosystem. Effective tagging ensures that products are easily searchable and discoverable and allows customers to find desired items swiftly and precisely. However, tags obtained through human manual annotation tend to be subjective and costly, often requiring significant time and resources [14]. As the scale of e-commerce platforms grows, so does the complexity of managing product information, prompting the need for advanced, automated solutions that ensure consistency, accuracy, and speed. Traditional methods, like rule-based tagging and keyword matching, rely on predefined heuristics, including regular expression matching, pattern-based tagging, and attribute extraction to assign tags [18]. These approaches standardize tags by mapping varied product descriptions to consistent labels through techniques like synonym mapping and phrase detection [17]. However, they require significant effort to design and maintain, making them less scalable as system complexity grows.

More recent approaches utilize large language models (LLMs) to capture semantic and contextual nuances from product descriptions, titles, and user reviews [9]. LLMs can discern detailed product attributes and generate tags that reflect both explicit and implicit features. Despite their strengths, traditional LLM-based tagging methods have notable limitations, particularly in capturing inter-product relationships and adapting to dynamic customer preferences. Prior approaches often require significant effort to aggregate product IDs along with textual information, imposing restrictions and demanding complex loss function designs, making real-time inference less feasible. This can result in tags that, while semantically accurate, may not fully align with evolving user needs or market trends.

To overcome these challenges, this paper proposes **Behavior-enhanced Item Tagging with Finetuned Large Language Models**, a novel framework that integrates user behavioral data into the model finetuning process, using LoRA in an unsupervised manner. The approach enables LLMs to generate more contextually relevant tags, named Behavior-enhanced Tags (BETags). The proposed methodology focuses on leveraging finetuned LLMs to enhance tagging accuracy and interpretability by integrating user data and establishing an automated tagging pipeline capable of supporting advanced recommendation systems. BETags are evaluated upon four downstream tasks, including different recommendation and

retrieval scenarios, using metrics such as Hit Rate (HR@k) and Normalized Discounted Cumulative Gain (NDCG@k).

In summary, this work makes the following main contributions:

- Our approach incorporates user behavioral data into the tagging process, allowing for the creation of contextually relevant tags that better align with real-world user preferences.
- Our method introduces a fully automated tagging pipeline that only requires session data. By integrating session data into the LLM's original causal task, our approach enables the model to naturally comprehend session knowledge without additional design efforts, offering a more efficient and adaptable solution that aligns with evolving user needs and market trends.
- Our BETags' effectiveness is validated through four different downstream tasks. Additionally, combining BETags with retrievers is able to achieve significantly lower time complexity while maintaining comparable performance to well-known recommendation models.

## 2 Related Works

### 2.1 Automatic Tagging System

Traditional tagging systems often rely on multi-label text classification models, which treat human-annotated tags as ground-truth labels and use item descriptions as input to predict these labels. These models have been widely adopted in conventional tagging frameworks, assuming that the human-defined tags fully capture the item's attributes. Binary Relevance (BR) [19] is a straightforward method that decomposes multi-label classification into multiple independent binary classification tasks. Classifier Chains (CC) [15] extend BR by arranging classifiers in a chain structure, where the prediction depends on the predictions of previous labels. ML-KNN [20] is an adaptation of the k-nearest neighbors algorithm for multi-label learning, calculating the probability of each label being relevant based on neighboring instances, making it suitable for small datasets. While effective in structured text categorization tasks, these models can be sub-optimal for real-world e-commerce scenarios, where item descriptions are sparse, vary in quality, or do not provide sufficient context for accurate tagging. Treating human-annotated tags as definitive ground truth without considering contextual factors or dynamic user preferences results in rigid tagging systems that may not adapt well to changing catalog features or evolving customer behaviors.

Using advanced language models for tagging, TagGPT [9] introduces a zero-shot multimodal tagging framework using prompt-based LLMs and unsupervised sentence embeddings to generate tags from diverse data automatically. This system introduces a post-processing module designed to improve the overall quality and efficiency of its tagging system. The module first removes tags that are too common or rare, as high-frequency tags may lack distinctiveness, and low-frequency tags may be insignificant. Next, it reduces the system's scale by fusing semantically similar tags, which is achieved by encoding all tags using an unsupervised pre-trained text encoder and calculating their cosine similarity.

In addition to handling noise and redundancy, the framework enables multimedia content tagging without requiring domain-specific training. However, while TagGPT effectively manages individual item information, it primarily focuses on single-item tagging and may overlook underlying item-item interactions that could be revealed through user browsing behaviors, potentially limiting its adaptability in dynamic environments.

### 2.2 LLM in Text-Related Tasks

Large language models (LLMs) are highly versatile, excelling at leveraging commonsense knowledge and reasoning. Researchers have explored their capabilities across a broad spectrum of text-related tasks, leveraging their proficiency in interpreting complex contexts, generating coherent narratives, and capturing subtle relationships. LLM-Rec [12] enhances item textual content by generating detailed tags and paraphrased descriptions, boosting even simple models with enriched text. TallRec [1] demonstrates how LLMs can be efficiently fine-tuned for domain-specific recommendation tasks using LoRA [7], maintaining effectiveness while reducing computational overhead. GPT4Rec [11] generates hypothetical search queries based on user history using multi-beam generation, improving relevance and diversity in recommendations. P5 [2] highlights the adaptability of LLMs by unifying different tasks into a text-based framework, allowing LLMs to handle multiple tasks with minimal fine-tuning. Overall, LLMs offer powerful capabilities for text generation and can easily adapt to varied scenarios through lightweight parameter-efficient finetuning, maintaining their rich understanding and reasoning skills. While traditional tagging methods provide a foundation for structured categorization, they often struggle to adapt to the dynamic and evolving nature of e-commerce. Several advanced models have made attempts to address these challenges, yet they fall short in capturing deeper user behavior patterns and item-item interactions. Our proposed method leverages the strengths of LLMs, known for their proficiency in text-related tasks, to create a tagging approach that overcomes these limitations, offering a more beneficial tagging solution.

## 3 Methodology

In this study, we propose BETag, a framework for an automated behavior-enhanced tagging system designed to generate enhanced tags by integrating user behavior data. Our goal is to generate a set of BETags that not only capture the semantic attributes of items but also align closely with user preferences and behaviors. As illustrated in Figure 1, BETag follows a multi-step process. First, we generate base tags by leveraging LLMs to produce initial semantic tags. Next, we apply behavior-enhanced finetuning to incorporate user behavioral data, followed by the final tag generation, where BETags are created based on both item characteristics and behavioral insights. The details of each process are explained in the following sections.

### 3.1 Problem Formulation

The core objective of our methodology is to develop an automated tagging system that leverages item context (e.g., title, description) and user behavior data to generate personalized, behavior-enhanced tags (BETags). BETags could be utilized in downstream tasks such

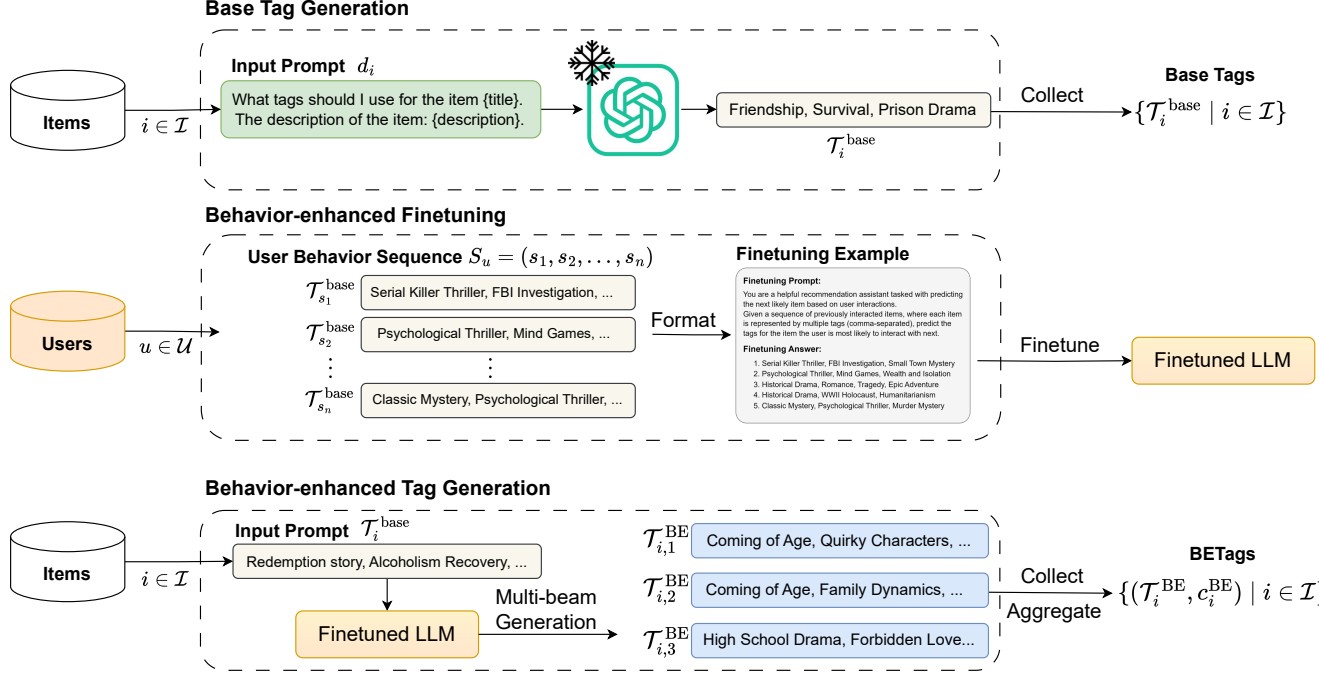

**Figure 1: Overall framework**

as recommendation systems and search optimization, enhancing the relevance and accuracy of these systems.

*3.1.1 User Historical Behavior Sequences.* Given an item set $\mathcal{I}$, each user $u \in \mathcal{U}$ is associated with a historical behavior sequence that captures the user's most recent $n$ interactions with items in chronological order. This sequence is denoted as $S = (s_1, s_2, \ldots, s_j, \ldots, s_n)$, where each $s_j \in \mathcal{I}$ represents the $j$-th item the user interacted with.

*3.1.2 Tagging System.* The tagging system for items is defined by $\{ (\mathcal{T}_i, c_i) \mid i \in \mathcal{I} \}$, where tags for each item are represented as a multiset $(\mathcal{T}_i, c_i)$. Here, $\mathcal{T}_i$ denotes the underlying set of distinct tags for item $i$ and $c_i : \mathcal{T}_i \rightarrow \mathbb{Z}_+$ is a function that indicates the number of occurrences of each tag $t$, with $c_i(t)$ representing the count of tag $t$.

## 3.2 Base Tag Generation

The purpose of base tag generation is to leverage an LLM's ability to understand and summarize item attributes into concise, generalizable tags that describe key product features. Base tag generation forms the first step in the BETag framework, where LLMs are employed to generate initial product tags. Specifically, tags are derived by prompting the LLM with item-specific information such as the product's title, description, and relevant metadata. Each item is processed independently, without regard for its relationship to other items or user-item interaction data.

We employ a straightforward prompting approach akin to that used by Lyu *et al.* [12]. The base tags of an item $i$ are obtained by:

$$\mathcal{T}_i^{\text{base}} = \text{LLM}\left(\text{prompt}^{\text{base}}(d_i), \epsilon\right), \qquad (1)$$

where $\text{prompt}^{\text{base}}(\cdot)$ denotes the template used to format the prompt using the descriptive content $d_i$, and $\text{LLM}(\cdot, \epsilon)$ denotes the LLM tag generation process with $\epsilon$ representing randomness introduced in the sampling processes for auto-regressive generation.

These base tags will be used for further refinement and personalization in subsequent steps. Note that the base tag generation is limited by its lack of contextualization across items since it only focuses on the item information itself. Consequently, while the base tags accurately capture the semantic characteristics of items, they lack the ability to generalize effectively across different contexts and fail to incorporate user behavior.

## 3.3 Behavior-enhanced Finetuning

To address the limitations of the base tag generation, where item-item interactions and user behavior are not considered, Behavior-enhanced finetuning refines the LLM by incorporating user behavioral data, improving the contextual relevance of item tags. User behavior sequences are formatted into finetuning examples, enabling the LLM to learn relationships between items based on users' interaction histories. This design allows the fine-tuned LLM to capture context-specific knowledge and generate behavior-enhanced tags that better reflect users' preferences and browsing patterns.

Specifically, given a user behavior sequence $S = (s_1, s_2, \ldots, s_n)$ from the training set, the base tags for each item in the sequence, $(\mathcal{T}_{s_1}^{\text{base}}, \mathcal{T}_{s_2}^{\text{base}}, \ldots, \mathcal{T}_{s_n}^{\text{base}})$, are extracted. These base tags represent simplified item descriptions, concatenated into a single string separated by commas for fine-tuning purposes. The tags for different items are arranged on separate lines in the order of the user's interactions, preserving the temporal order.

For clarity, a finetuning example is illustrated in Figure 2, where a sequence of five items, each with its corresponding set of base tags, is transformed into the structured format used for finetuning the LLM. We finetune the LLM with its parameters denoted by $\Phi$,

---

**Finetuning Prompt:**

You are a helpful recommendation assistant tasked with predicting the next likely item based on user interactions.
Given a sequence of previously interacted items, where each item is represented by multiple tags (comma-separated), predict the tags for the item the user is most likely to interact with next.

**Finetuning Answer:**

1. Serial Killer Thriller, FBI Investigation, Small Town Mystery
2. Psychological Thriller, Mind Games, Wealth and Isolation
3. Historical Drama, Romance, Tragedy, Epic Adventure
4. Historical Drama, WWII Holocaust, Humanitarianism
5. Classic Mystery, Psychological Thriller, Murder Mystery

---

**Figure 2: An example of behavior-enhanced finetuning on Movielens-1M**

using the following objective:

$$\Phi^* = \arg\max_{\Phi} \sum_{y \in \mathcal{D}_{\text{train}}} \sum_{t=1}^{|y|} \log P_\Phi(y_t \mid x, y_{<t}), \tag{2}$$

where $\mathcal{D}_{\text{train}}$ denotes the training set, and $y$ represent the output tokens, with $y_t$ being the $t$-th token and $y_{<t}$ denoting the sequence of tokens preceding the $t$-th token. In our case, $x$ represents the input prompt, which is a static task-guiding instruction, as illustrated in Figure 2. We observed that including this instruction helped the LLM converge more efficiently, requiring fewer training epochs. The output $y$ is tokenized from the finetuning example derived from a user behavior sequence, as described earlier and depicted in Figure 2.

The probability distribution $P_\Phi(y_t \mid \cdot)$ is parameterized by the LLM, where $\Phi$ represents its parameters. Only a small portion of $\Phi$ is updated during finetuning using parameter-efficient techniques, LoRA [7], as outlined in other methods finetuning LLM for domain-specific adaptation for recommendation [12, 21].

The structured format used for finetuning allows the model to learn item-item interactions based on how items are encountered in the user's behavior sequence. As a result, the LLM can generate more context-aware and behavior-enhanced tags during inference, improving the personalization of item tags.

## 3.4 BETag Generation

The *BETag Generation* phase outlines the process of generating product tags that capture item-item interactions. This step builds on the initial *base tags* by incorporating additional user behavior history to create more sophisticated and interconnected tags. The process involves the behavior-enhanced finetuned LLM to predict tags for the next item in a user's behavior history. The BETags are designed to reflect relationships between products, which helps identify cross-selling opportunities and improve search and navigation efficiency. This method aims to provide high-quality tags that not only enhance the accuracy of recommendations but also address the limitations of the base tags by leveraging the broader context of item interactions and user behaviors.

To generate BETags for a given item, the item's base tags are used as the input prompt to the finetuned LLM. These base tags, which serve as simple descriptors, are concatenated into a single string, with each tag separated by commas or other appropriate delimiters. This structured prompt is then provided to the model for inference.

In contrast to the finetuning stage, where user-item interaction sequences are considered, the BETag generation process focuses solely on individual items. The finetuned model utilizes its learned understanding of item relationships to generate behavior-enhanced tags without the presence of user behavior sequences. As illustrated in Figure 3, the input consists of the item's base tags, ensuring that the generated BETags capture both the item's inherent characteristics and its potential associations with other items in a broader context. Note that the second line is left blank since we only focus on the considered item, and no user behavioral data is used explicitly in the BETag generation phase.

Inspired by Li *et al.* [11], we employ multi-beam generation to obtain BETags. Specifically, for each beam $b \in \{1, \cdots, m\}$, the tags are generated as:

$$\mathcal{T}_{i,b}^{\text{BE}} = \text{LLM}_{\Phi^*}\left(\text{prompt}^{\text{BE}}\left(\mathcal{T}_i^{\text{base}}\right), \epsilon_b\right), \tag{3}$$

where $\text{prompt}^{\text{BE}}(\cdot)$ denotes the template used to format the prompt, and $\text{LLM}_{\Phi^*}(\cdot, \epsilon_b)$ denotes the tag generation process using the behavior-finetuned LLM with $\epsilon_b$ representing randomness introduced in the sampling processes for the $b$-th beam.

The final BETags $(\mathcal{T}_i^{\text{BE}}, c_i^{\text{BE}})$ for item $i$ are then aggregated from the tags generated in each beam, where

$$\mathcal{T}_i^{\text{BE}} = \bigcup_{b=1}^{m} \mathcal{T}_{i,b}^{\text{BE}}, \tag{4}$$

$$c_i^{\text{BE}}(t) = \sum_{b=1}^{m} \left[\!\left[ t \in \mathcal{T}_{i,b}^{\text{BE}} \right]\!\right], \tag{5}$$

$[\![\cdot]\!]$ denotes an indicator function and $m$, the number of beams for behavior-enhanced tag generation, is treated as a hyperparameter.

As illustrated in Figure 3, the generation for each beam is terminated upon producing a newline character. Increasing the number of beams allows for the generation of a greater number of tags per item, thereby enhancing the overall tag diversity.

## 3.5 Downstream Applications

The BETags can be used as supplementary information for various downstream tasks, such as recommendation systems, item classification, or search optimization.

To assess the effectiveness of our tagging methodology, we conduct experiments across multiple retrieval and recommendation-related tasks in diverse real-world application scenarios. Each task reflects a distinct combination of user- or item-based paradigms, focusing on either retrieval or recommendation applications. Our goal is to evaluate whether integrating BETags into existing retrieval/recommendation models enhances performance compared to using native tags (e.g., human-annotated or genre-based) and other automated tagging systems.

> **Generation Prompt:**
> You are a helpful recommendation assistant tasked with ...
>
> 1. Serial Killer Thriller, FBI Investigation, Small Town Mystery
> 2.
>
> **BETags #1:**
> Psychological Thriller, Mystery, Mind Games, Hannibal Lecter
> **BETags #2:**
> Crime Drama, Criminal Minds, Hannibal Lecter
> **BETags #3:**
> Action-Packed, Post-Apocalyptic Future, Survival Thriller

**Figure 3: An example of behavior-enhanced tag generation on Movielens-1M**

*3.5.1 Item-based Retrieval.* In item-based retrieval tasks, the system aims to retrieve the next relevant item by using the tags associated with the current item as a query. Traditional retrieval systems typically handle tags as terms within a document, representing the item, and perform tag-query matching to retrieve relevant items. This method is computationally efficient, offering low latency and minimal resource consumption, which makes it well-suited for real-time applications.

Formally, given a tagging system, as defined in Section 3.1.2, retrievers rank items based on a query $Q = (\mathcal{T}_i, c_i)$ composed of tags for the current item $i$. The ranking function $R$ representing the retriever generally takes the form:

$$\hat{\mathbf{y}} = R(\mathcal{D}, Q), \qquad (6)$$

where $\mathcal{D} = \{ (\mathcal{T}_{i'}, c_{i'}) \mid i' \in \mathcal{I} \}$ are the items to be retrieved, represented by their tags, and $\hat{\mathbf{y}} \in \mathbb{R}^{|\mathcal{I}|}$ denotes the predicted ranking scores for items.

For the retriever $R$, we experiment with BM25 [16], and BiRank [4]. Details on how the ranking scores are calculated are provided in Appendix A. Additionally, as discussed in Appendix A, the retriever functions along with a given tagging system $R(\mathcal{D}, \cdot)$ can be formulated as a linear method and can be precomputed offline. This feature, combined with the BETag tagging system, makes the retrieval application scenarios highly efficient in terms of latency and computational resources.

*3.5.2 User-based Retrieval.* In user-based retrieval tasks, the objective is to retrieve the next item based on a user's behavior history. The tags from items the user has interacted with are aggregated to form a query. Specifically, the query $Q^{\text{user}} = (\mathcal{T}^{\text{user}}, c^{\text{user}})$ is constructed by aggregating the tags of items from the user's behavior sequence $S = (s_1, s_2, \ldots, s_n)$, defined as:

$$\mathcal{T}^{\text{user}} = \bigcup_{j=1}^{n} \mathcal{T}_{s_j}, \qquad (7)$$

$$c^{\text{user}}(t) = \sum_{j=1}^{n} w_j \tilde{c}_{s_j}(t), \qquad (8)$$

where $\tilde{c}_{s_j} : \mathcal{T}^{\text{user}} \to \mathbb{Z}_+$ with $\tilde{c}_{s_j}(t) = c_{s_j}(t)$ if $t \in \mathcal{T}_{s_j}$, $\tilde{c}_{s_j}(t') = 0$ if $t' \notin \mathcal{T}_{s_j}$. Here $w_i$, representing the interaction weights, can be either uniform, $w_i^{\text{uniform}} = 1/n$, or linear, $w_i^{\text{linear}} = \frac{2i}{n(n-1)}$. The

weighting scheme is treated as a hyperparameter, selected based on model performance.

The next item is then retrieved based on the score function:

$$\hat{\mathbf{y}} = R(\mathcal{D}, Q^{\text{user}}), \qquad (9)$$

where $\mathcal{D} = \{ (\mathcal{T}_{i'}, c_{i'}) \mid i' \in \mathcal{I} \}$ are the tag-represented items to be retrieved.

*3.5.3 User-based Recommendation.* Recommender systems that learn item ID embeddings have demonstrated robust performance across various recommendation tasks. Incorporating textual embeddings from pre-trained encoders further enhances their ability to generalize and improve the quality of recommendations. In this context, BETags provide augmented textual information, enriching item embeddings and improving the overall recommendation process by introducing behavior-enhanced, semantically rich tags.

To leverage BETags effectively in this framework, we assign the generated BETags as the textual features for each item. We first select the top-$k$ tags $\tilde{\mathcal{T}}_i$ based on their occurrences as:

$$\tilde{\mathcal{T}}_i = \arg \max_{\mathcal{T}' \subseteq \mathcal{T}, |\mathcal{T}'|=k} \sum_{t \in \mathcal{T}'} c_i(t), \qquad (10)$$

The textual item embedding for an item $i$ is derived from tags $\tilde{\mathcal{T}}_i$ using a pretrained language model and is further combined with a learnable ID embedding as its item embedding.

*3.5.4 Item-based Recommendation.* For item-based recommendations, our objective is to predict the next item based on the current item. We employ a sliding window approach on the existing user behavior sequences in the training set. By creating pseudo users with only one interacted item, we can train the recommender to leverage the context of the current item for predicting subsequent items.

## 4 Experiments

In this section, we conduct experiments to answer the following research questions:

- **RQ1**: How do the BETags improve the performance of downstream user- and item-based tasks in recommendation and retrieval systems?
- **RQ2**: To what extent do the LLM finetuning and multi-beam generation contribute individually to the performance improvements observed in downstream tasks when using BETags?
- **RQ3**: How does increasing the number of beams in multi-beam generation impact the performance, and how is this improvement linked to changes in tag distribution?
- **RQ4**: How does BETag address the item-tag sparsity issue typically encountered in tags generated by LLMs?

### 4.1 Experiment Setup

*Datasets.* The tasks were conducted on three real-world datasets, *FreshFood*, *MovieLens-1M*, and *Amazon (Scientific)*. *FreshFood* is a session-based dataset derived from a Taiwanese e-commerce platform for fresh food, where sessions are treated as users to align with the other datasets. The interaction type is user browsing history, with native tags represented by human-annotated keywords for

**Table 1: Statistics of the datasets. "Repeatable" indicates whether the dataset allows repeated interactions between users and items.**

|  | Scientific | Movielens-1M | FreshFood |
|---|---|---|---|
| #Users | 11,041 | 5,954 | 27,059 |
| #Items | 5,327 | 2,803 | 357 |
| #Inters. | 76,896 | 985,332 | 167,176 |
| #Inters./user | 6.96 | 165.49 | 6.18 |
| #Inters./item | 14.44 | 351.53 | 468.28 |
| User-Item Sparsity | 98.88% | 94.10% | 98.83% |
| Repeatable | True | False | True |

each item. *MovieLens-1M* is a well-known public dataset consisting of movie reviews, with native tags including genres and human-annotated keywords; plot summaries, obtained from IMDb, are used as item descriptions. Finally, *Amazon (Scientific)* contains product reviews in the scientific category, and the items are represented by their product descriptions without any native tags available.

*Baselines.* The evaluation compares different tagging systems, with the baselines consisting of native tags from each dataset and TagGPT [9], which utilizes an LLM for automatic tagging followed by post-processing steps, including frequency filtering and semantic fusion. For retrieval tasks, we include a *Popular* baseline, a common strategy used in e-commerce that ranks items based on popularity. In recommendation tasks, we use SASRec [8], a well-known and effective ID-based recommender, as the baseline model.

*Data Splitting Strategy.* Following previous work by Hou *et al.* [6], we employ the widely used leave-one-last-item strategy for data splitting, where the last item for each user is used for testing, the second-to-last item for validation, and the remainder for training. To prevent data leakage, the LLM finetuning for the tagging system and the downstream recommendation modules use the same training set.

*Evaluation Settings.* We use two standard Top-N evaluation metrics, Hit Rate@10, and NDCG@10, to assess recommendation performance [3, 5], later denoted as H@10 and N@10 for simplicity. For user-based tasks, we follow established practices [6, 8]. If user interactions are non-repeatable, each positive item is paired with 100 negative items that the user has not interacted with. In repeatable cases, 100 negative items are randomly sampled from the overall item pool. The repeatability of each dataset is detailed in Table 1. For item-based tasks, 100 negative items are randomly selected from the item pool for each evaluation.

We evaluate our BETag in two scenarios: using retrievers and recommenders, each tailored for tasks including item-based and user-based recommendations.

*Retriever Modules.* For retrievers, we employ BM25 [16] and BiRank [4], both of which rely solely on textual information without using item IDs, meaning user-item interactions are not directly considered. BiRank optimizes ranking in bipartite graphs by leveraging item-text relations for graph construction. It iteratively updates ranking scores until convergence, suitable for item retrieval. BM25, a widely used ranking function in information retrieval, estimates document relevance through term frequency and inverse document

frequency (TF-IDF). It adjusts for term saturation and document length, offering efficiency in ranking large text collections. More details for how these two retrievers are integrated into the retrieval process can be found in Appendix A.

*Recommendation Modules.* For the recommender, we select the transductive settings of UniSRec [6] as our recommender, which is built on SASRec but utilizes the associated description text of items to learn transferable representations across various recommendation scenarios, with tags serving as augmented descriptions for each item.

## 4.2 Main Results (RQ1)

*4.2.1 Retrieval Tasks.* The performance of BETag in both user-based and item-based retrieval tasks consistently surpassed Native tags and TagGPT, shown in Table 2.

In the user-based task, where BM25 and BiRank rely on user behavior sequence, BETag achieved the highest H@10 and N@10 scores, demonstrating better alignment with user preferences. Similarly, in the item-based task, which involves less input information and thus lower overall performance, BETag continued to outperform both native tags and TagGPT across both models. Additionally, the integration of BETag with retrievers like BM25 and BiRank offers significant advantages in terms of latency and computational efficiency, as these models can leverage offline-generated tags, reducing real-time computational overhead while maintaining high retrieval accuracy.

*4.2.2 Recommendation Tasks.* Table 3 presents the results for the baseline model SASRec, as well as versions of the UniSRec model utilizing different text inputs: Native Tags, TagGPT, and the proposed BETag. The comparison between these models shows clear trends in performance. SASRec generally provides solid results, especially on the Movielens-1M dataset, but is often outperformed by the other models in specific scenarios. UniSRec with Native Tags performs well, particularly in user-based recommendations for Movielens-1M, but struggles somewhat on other datasets.

However, the standout performer across all approaches is using BETag with UniSRec. BETag consistently achieves the highest scores in both H@10 and N@10 metrics, especially on the Scientific dataset, where it shows a significant improvement over other models. This superior performance indicates that our proposed approach to tagging and recommendation significantly enhances the model's ability to recommend relevant items. In particular, BETag's robust performance on both user-based and item-based recommendations highlights its versatility and the effectiveness of its tagging mechanism in improving recommendation quality.

In summary, while SASRec and UniSRec (with Native Tags and TagGPT) demonstrate strong results, especially in certain datasets and metrics, the proposed BETag method consistently surpasses them, particularly in complex recommendation tasks such as those presented by the Scientific dataset. This suggests that BETag offers a more reliable and precise approach to generating high-quality recommendations in various domains.

**Table 2: Retrieval task performance**

| | User-Based | | | | | | Item-Based | | | | | |
|---|---|---|---|---|---|---|---|---|---|---|---|---|
| | Scientific | | Movielens-1M | | FreshFood | | Scientific | | Movielens-1M | | FreshFood | |
| | H@10 | N@10 | H@10 | N@10 | H@10 | N@10 | H@10 | N@10 | H@10 | N@10 | H@10 | N@10 |
| **Popular** | 0.2871 | 0.1672 | 0.3866 | 0.2115 | 0.4158 | 0.2292 | 0.2871 | 0.1672 | 0.3467 | 0.1784 | 0.4158 | 0.2292 |
| **BM25** | | | | | | | | | | | | |
| - Native Tags | - | - | 0.3488 | 0.1910 | 0.5145 | 0.3216 | - | - | 0.3179 | 0.1718 | 0.3990 | 0.2361 |
| - TagGPT | 0.3430 | 0.2241 | 0.2929 | 0.1601 | 0.4666 | 0.3165 | 0.2862 | 0.1930 | 0.2104 | 0.1116 | 0.2928 | 0.1896 |
| - BETag (Ours) | **0.4883** | **0.3098** | **0.4723** | **0.2791** | **0.5152** | **0.3350** | **0.4249** | **0.2747** | **0.3779** | **0.2160** | **0.4603** | **0.2712** |
| **BiRank** | | | | | | | | | | | | |
| - Native Tags | - | - | 0.3939 | 0.2200 | 0.5287 | 0.3493 | - | - | 0.2927 | 0.1563 | 0.4062 | 0.2432 |
| - TagGPT | 0.3741 | 0.2409 | 0.3095 | 0.1663 | 0.4832 | 0.3284 | 0.3040 | 0.2024 | 0.2244 | 0.1170 | 0.3260 | 0.2026 |
| - BETag (Ours) | **0.4681** | **0.2991** | **0.4667** | **0.2732** | **0.5526** | **0.3569** | **0.4239** | **0.2735** | **0.3764** | **0.2176** | **0.4791** | **0.2779** |

**Table 3: Recommendation task performance**

| | User-Based | | | | | | Item-Based | | | | | |
|---|---|---|---|---|---|---|---|---|---|---|---|---|
| | Scientific | | Movielens-1M | | FreshFood | | Scientific | | Movielens-1M | | FreshFood | |
| | H@10 | N@10 | H@10 | N@10 | H@10 | N@10 | H@10 | N@10 | H@10 | N@10 | H@10 | N@10 |
| **SASRec** | 0.5057 | 0.3342 | 0.7335 | 0.4904 | 0.5698 | 0.3399 | 0.3930 | 0.2409 | 0.6636 | 0.4304 | 0.5396 | 0.3157 |
| **UniSRec** | | | | | | | | | | | | |
| - Native Tags | - | - | 0.7462 | 0.5039 | 0.5689 | 0.3271 | - | - | 0.6634 | 0.4244 | 0.4680 | 0.2639 |
| - TagGPT | 0.5308 | 0.3400 | 0.7474 | 0.5066 | 0.5618 | 0.3254 | 0.4415 | 0.2754 | 0.6547 | 0.4254 | 0.5168 | 0.3041 |
| - BETag (Ours) | **0.5801** | **0.3742** | **0.7523** | **0.5106** | **0.5741** | **0.3417** | **0.4740** | **0.2972** | **0.6703** | **0.4344** | **0.5590** | **0.3245** |

## 4.3 Discussion and Analysis

*4.3.1 Evaluating the Contribution of LLM Finetuning and Multi-Beam Generation to BETag Performance (**RQ2**).* To address **RQ2**, we investigate the individual contributions of LLM finetuning and multi-beam generation to the performance of BETags on item-based retrieval tasks. We conduct experiments with four variations: the full BETag method, without LLM finetuning, without multi-beam generation, and without both, examining the extent of performance degradation when these components are excluded. The results are summarized in Table 4.

First, when we omit LLM finetuning, tags are generated using a non-behaviorally finetuned LLM through multiple generations.This setup isolates the impact of multi-beam generation, with notable performance drops in retrieval tasks using BM25 and BiRank. Interestingly, datasets with lower user-item sparsity (where more behavioral data is available for finetuning) show a larger decrease in performance, underscoring the importance of behavior-driven LLM finetuning.

Conversely, when we exclude multi-beam generation, tags are generated with a behaviorally finetuned LLM using single-beam generation. This setting isolates the contribution of finetuning. Similar to the results without finetuning, performance drops are observed across BM25 and BiRank.

Finally, when both LLM finetuning and multi-beam generation are excluded, performance drops even further, demonstrating that both components significantly enhance the base tags generated in the initial step of BETag. This finding highlights that either LLM finetuning or multi-beam generation alone leads to a marked improvement over naive base tags, but the combination of both yields

**Table 4: Evaluation of LLM finetuning and multi-beam generation in BETag Performance on item-based retrieval tasks**

| | Scientific | | Movielens-1M | | FreshFood | |
|---|---|---|---|---|---|---|
| | H@10 | N@10 | H@10 | N@10 | H@10 | N@10 |
| **BM25** | | | | | | |
| - BETag (Ours) | **0.4249** | **0.2747** | **0.3779** | **0.2160** | **0.4603** | **0.2712** |
| - w/o Finetuning | 0.3582 | 0.2562 | 0.2382 | 0.1288 | 0.2918 | 0.1898 |
| - w/o Multi-beam | 0.2234 | 0.1508 | 0.2093 | 0.1138 | 0.3493 | 0.2153 |
| - w/o Both | 0.1570 | 0.1064 | 0.1740 | 0.0848 | 0.2270 | 0.1437 |
| **BiRank** | | | | | | |
| - BETag (Ours) | **0.4239** | **0.2735** | **0.3764** | **0.2176** | **0.4791** | **0.2779** |
| - w/o Finetuning | 0.3907 | 0.2723 | 0.2430 | 0.1309 | 0.3210 | 0.2001 |
| - w/o Multi-beam | 0.2863 | 0.1821 | 0.2277 | 0.1203 | 0.3693 | 0.2213 |
| - w/o Both | 0.2059 | 0.1340 | 0.1426 | 0.0744 | 0.2432 | 0.1564 |

the best results. Similar results for user-based recommendations are also shown in Appendix B.

*4.3.2 Exploring the Effect of Beam Count on BETag Performance and Diversity (**RQ3**).* To gain further insight into the impact of multi-beam generation on BETag, we analyze its influence on item-based retrieval tasks and tag distribution properties.

Figure 4 illustrates the performance (N@10) over the number of beams used in generating BETags, with two curves representing BM25 and BiRank, respectively. As the number of beams increases, we observe a significant rise in performance for both retrievers, with the improvement starting to saturate around 10 beams.

In Figure 5, we examine how multi-beam generation affects the distribution of item tags using the FreshFood dataset. Specifically, we plot the occurrences vector $A_i$ of an item's tags over different numbers of beams, excluding zero entries for clarity. The results show that as the number of beams increases, the tag distribution stabilizes. Without multi-beam generation, some critical tags, which appear frequently with a higher number of beams, are absent. Furthermore, the number of tags with non-zero occurrences grows with more beams, indicating an increase in tag diversity. This growth reflects a richer and more varied set of item-tag connections, alleviating the item-tag sparsity issue commonly observed in LLM-generated tagging systems.

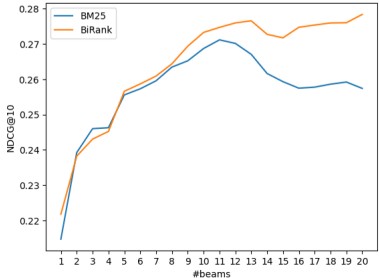

**Figure 4: N@10 over #beams in item-based retrieval**

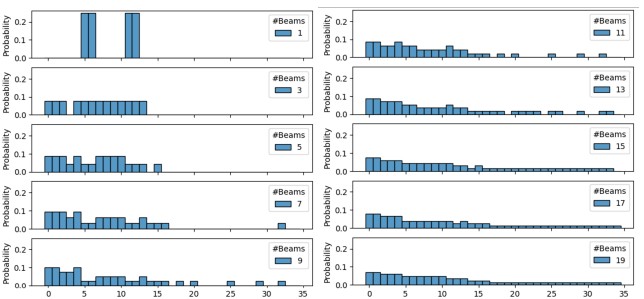

**Figure 5: Tag distributions for an item in the FreshFood**

*4.3.3 Reducing Item-Tag Sparsity in Automated Product Tagging (RQ4).* The issue of item-tag sparsity is a significant challenge in the automatic generation of tags using LLMs. LLM-generated tags, while effective in capturing semantic meaning, often produce a large proportion of tags that are attached to only a single item, limiting their applicability in many practical scenarios. For instance, in movie tagging, semantically similar phrases like "Teenage rebellion," "Youth rebellion," and "Youthful rebellion" may be treated as distinct, leading to a fragmented tag space.

To mitigate this issue, previous methods such as TagGPT have employed postprocessing techniques, including frequency filtering and semantic fusion. Semantic fusion groups tags with close meanings by applying a threshold on the cosine similarity of their embeddings.

Our BETag approach also addresses the item-tag sparsity issue. Figure 6 presents a comparison of the tag popularity distribution

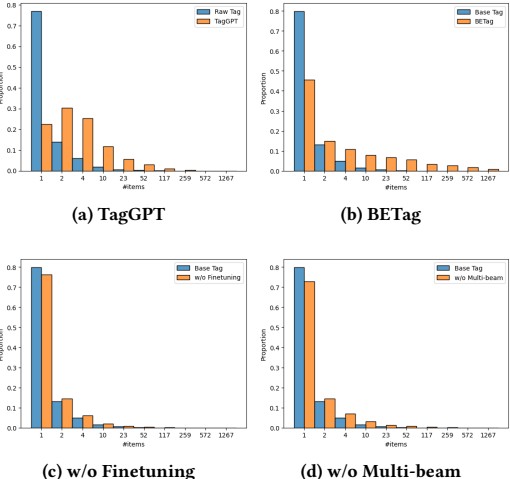

**Figure 6: Popularity on MovieLens-1M**

[9] in the MovieLens-1M dataset. In Figure 6a, we compare the tag distribution between TagGPT's postprocessed tags and raw LLM-generated tags (denoted "Raw Tag"). Similarly, Figure 6b contrasts the popularity distribution of BETag against Raw Tags. This shows that BETag effectively mitigates item-tag sparsity, resulting in a more balanced distribution of tags across items.

This improvement is driven by two core components: behavior finetuning and multi-beam generation. Figure 6c shows the tag distribution when comparing BETag without behavior finetuning to Raw Tags. Multi-beam generation generates more tags per item, thereby covering a wider range of item attributes, leading to more tags being shared among different items. Figure 6d presents the comparison between BETag without multi-beam generation and Raw Tags, where finetuning plays a crucial role in reducing the chance of generating distinct unseen tags, resulting in the smoothing effect. These two mechanisms collectively help BETag address item-tag sparsity by expanding tag coverage while maintaining precision.

## 5 Conclusion

We present BETag, a novel automated tagging pipeline that integrates real-world user behavior data to generate personalized, behavior-enhanced tags. By addressing item-tag sparsity issues in LLM-based tagging methods, BETag enhances tags' accuracy and contextual relevance, aligning them more closely with user preferences. We have successfully validated BETag with three different datasets and two common tasks: retrieval and recommendation. BETag enhances the performance of downstream tasks, surpassing state-of-the-art models while significantly reducing computational complexity for real-world applications. It has demonstrated its effectiveness in improving search relevance, product discoverability, and personalized recommendations. Its scalability and efficiency make it a practical solution for e-commerce platforms, enhancing user experiences and system performance without imposing real-time computational burdens.

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

## A  Retrievers

Given a set of documents $\mathcal{D} = \{ (\mathcal{T}_i, c_i) \mid i \in \mathcal{I} \}$ are the items to be retrieved, represented by their tags, the overall domain for tags across all item is identified as:

$$\mathcal{T} = \bigcup_{(\mathcal{T}_i, c_i) \in \mathcal{D}} \mathcal{T}_i. \tag{11}$$

A query used to retrieve the item is denoted by $Q = (\mathcal{T}^q, c^q)$ that consists of tags in the overall tag domain, where $\mathcal{T}^q \subseteq \mathcal{T}$ and $c^q : \mathcal{T}^q \to \mathbb{Z}_+$ denotes the occurrences, namely weights of each tag in the query.

The ranking function $R$ representing the retriever generally takes the form:

$$\hat{\mathbf{y}} = R(\mathcal{D}, Q), \tag{12}$$

where $\hat{\mathbf{y}} \in \mathbb{R}^{|\mathcal{I}|}$ denotes the predicted ranking scores for items.

Given the item set $\mathcal{I} = \{ i_1, i_2, \ldots, i_{|\mathcal{I}|} \}$ and overall tag domain $\mathcal{T} = \{ t_1, t_2, \ldots, t_{|\mathcal{T}|} \}$, we first vectorize the given documents as matrix $A \in \mathbb{R}^{|\mathcal{I}| \times |\mathcal{T}|}$ and the query as vector $\mathbf{q} \in \mathbb{R}^{|\mathcal{T}|}$, where

$$A_{j,k} = c_{i_j}(t_k), \tag{13}$$

$$\mathbf{q}_k = c^q(t_k). \tag{14}$$

We are going to show the ranking functions, no matter BM25 or BiRank, can be formulated as a linear method following the form:

$$\hat{\mathbf{y}} = W^R(A)\mathbf{q}, \tag{15}$$

where $W^R : \mathbb{R}^{|\mathcal{I}| \times |\mathcal{T}|} \to \mathbb{R}^{|\mathcal{I}| \times |\mathcal{T}|}$ and $\mathbf{q} \in \mathbb{R}^{|\mathcal{T}|}$.

With this property, given the tagging system as documents, we can always compute $W^R(A)$ offline. Furthermore, this format allows the methods to be easily parallelized with GPU, making the online recommendation extremely efficient.

### A.1  BM25

BM25 [16], one of the most widely used baseline search engines, is adopted as one of our retrieval model for the downstream recommendation tasks.

The BM25 model can be represented by a linear matrix $W^{\text{BM25}} \in \mathbb{R}^{|\mathcal{I}| \times |\mathcal{T}|}$ given documents $\{ (\mathcal{T}_i, c_i) \mid i \in \mathcal{I} \}$ with item set $\mathcal{I} = \{ i_1, i_2, \ldots, |\mathcal{I}| \}$ and overall tag domain $\mathcal{T} = \{ t_1, t_2, \ldots, t_{|\mathcal{T}|} \}$:

$$W_{j,k}^{\text{BM25}} = \text{IDF}(t_k) \cdot \frac{c_{i_j}(t_k) \cdot (k_1 + 1)}{c_{i_j}(t_k) + k_1 \left(1 - b + b \frac{\sum_{t \in \mathcal{T}_i} c_{i_j}(t)}{\frac{1}{|\mathcal{I}|} \sum_{i' \in \mathcal{I}} \sum_{t' \in \mathcal{T}_{i'}} c_{i'}(t')}\right)}, \tag{16}$$

where $k_1, b$ are free parameters account for the saturation of term frequency and the document length. Here, the IDF($\cdot$) is the inverse document frequency, given as:

$$\text{IDF}(t) = \ln\left(\frac{|\mathcal{I}| - N(t) + 0.5}{N(t) + 0.5} + 1\right), \quad (17)$$

$$N(t) = \sum_{i \in \mathcal{I}} [\![t \in \mathcal{T}_i]\!], \quad (18)$$

where $N : \mathcal{T} \to \mathbb{Z}_+$ denotes the number of items that contains tag $t$. The ranking scores of all items, denoted by $\hat{\mathbf{y}} \in \mathbb{R}^{|\mathcal{I}|}$, is then calculated as:

$$\hat{\mathbf{y}} = W^{\text{BM25}}\mathbf{q}. \quad (19)$$

## A.2 BiRank

BiRank [4] is a random-walk based algorithm similar to PageRank [13] but addresses the ranking problem on bipartite graphs. BiRank can be applied to general ranking scenario by constructing bipartite graph and task-specific query as the initial ranking scores for vertices, followed by BiRank iteration until convergence. The authors also further extended BiRank to rank on user-item-aspect tripartite graphs as *TriRank* for personalized recommendation ranking.

To align with other retrieval models such as BM25, we adopt the BiRank setting on item-aspect, i.e. item-tag, bipartite graph for our downstream recommendation tasks.

Given the documents $\mathcal{D} = \{ (\mathcal{T}_i, c_i) \mid i \in \mathcal{I} \}$ with item set $\mathcal{I} = \{ i_1, i_2, \ldots, |\mathcal{I}| \}$ and the overall tag domain $\mathcal{T} = \{ t_1, t_2, \ldots, t_{|\mathcal{T}|} \}$, the bipartite graph is constructed as $\mathcal{G} = (\mathcal{I} \cup \mathcal{T}, \mathcal{E})$. Here $\mathcal{I}$ and $\mathcal{T}$ become the vertex sets of items and tags respectively, and $\mathcal{E} = \{ (i, t) \mid t \in \mathcal{T}_i, i \in \mathcal{I} \}$ represents the edge set. $A \in \mathbb{R}^{|\mathcal{I}| \times |\mathcal{T}|}$, vectorized matrix representing the documents, is then used to denote weighted adjacency of the graph.

The corresponding BiRank transition matrix is the symmetric normalization of the adjacency:

$$S = D_i^{-1/2} A D_t^{-1/2}, \quad (20)$$

where $D_i, D_t$ are the diagonal matrices denoting the weighted degrees of item nodes, tag nodes respectively.

The iterative ranking process with $\mathbf{q}$ as the query for tag nodes and zero vector for item nodes can be obtained as:

$$\hat{\mathbf{y}} = \alpha(1 - \beta)\left(\sum_{i=0}^{\infty}(\alpha\beta SS^{\top})^i\right)S\mathbf{q}, \quad (21)$$

where $\alpha, \beta$ are two hyperparameters of BiRank.

## B  More Discussions

*Evaluating the Contribution of LLM Finetuning and Multi-Beam Generation to BETag Performance for User-based Recommendations.* Using BETag for the recommender consistently shows the best results across all datasets and metrics, confirming that the combination of finetuning and multi-beam search improves recommendation performance (as detailed in Table 5). The drop in performance when either component is removed highlights the importance of each. When fine-tuning is excluded, the performance decreases across all datasets, particularly on Movielens-1M. This suggests that finetuning helps the model adapt to the datasets, enhancing the accuracy of its recommendations. Similarly, the removal of

**Table 5: Evaluation of LLM finetuning and multi-beam generation in BETag Performance on user-based recommendations**

|  | Scientific | | Movielens-1M | | FreshFood | |
|---|---|---|---|---|---|---|
|  | N@10 | H@10 | N@10 | H@10 | N@10 | H@10 |
| **UniSRec** | | | | | | |
| - **BETag (Ours)** | **0.5801** | **0.3742** | **0.7523** | **0.5106** | **0.5741** | **0.3417** |
| - w/o Finetuning | 0.5769 | 0.3715 | 0.7386 | 0.4881 | 0.5625 | 0.3355 |
| - w/o Multi-beam | 0.5712 | 0.3692 | 0.7484 | 0.5004 | 0.5571 | 0.3250 |
| - w/o Both | 0.5623 | 0.3653 | 0.7309 | 0.4899 | 0.5539 | 0.3215 |

multi-beam search also leads to noticeable declines in performance, especially for the FreshFood dataset. This indicates that multi-beam search likely plays a crucial role in refining the model's ability to explore multiple potential outcomes before selecting the best recommendation, thus improving the overall hit rate and ranking performance.

The third ablated version, without both finetuning and multi-beam search, performs the worst in every case, further underscoring the importance of both techniques working together to maximize the model's effectiveness. This shows that each contributes independently to the model's performance, but their combination is crucial for achieving optimal results.

In conclusion, the ablation study demonstrates that both finetuning and multi-beam search are key components of the BETag model, each contributing significantly to its strong recommendation performance across various datasets. Their combined effect is most evident in the model's superior ranking and hit rate results, particularly on complex datasets like Movielens-1M and Scientific. Without these components, the model's ability to recommend relevant items effectively is notably reduced.

## C  Case Study

To illustrate the qualitative performance of BETag, we present a comparative case study using two examples from the MovieLens-1M dataset (*Tender Mercies* and *The Godfather*) and two examples from the Amazon (Scientific) dataset (*Sanding Roll* and *PLA Filament*). As shown in Table 6, we compare the human-annotated native tags, the LLM-generated base tags, and the behavior-enhanced BETags.

For the MovieLens-1M dataset, native tags for these films, composed of human-annotated genre labels (e.g., "Drama", "Romance"), tend to be concise but limited in their semantic depth. In contrast, LLM-generated base tags, though still sparse, provide richer semantic associations, capturing aspects such as "musical drama" or "family dynamics," which offer an accurate understanding of the film's themes. BETags further expand on these associations, incorporating broader semantic elements. For example, *Tender Mercies* is tagged with "Coming of Age," reflecting personal growth and transformation that align with the film's core narrative, even though such themes may not be immediately apparent from the movie's plot alone.

Further, we explore two items from the Amazon (Scientific) dataset: *Sanding Roll* and *PLA Filament*. In this dataset, the item information is more limited, consisting primarily of titles, brands, and categories. As a result, the generated base tags experience even

greater sparsity due to the nature of the dataset. Nevertheless, these base tags accurately capture the items' semantic properties based on the available information. Notably, for the "Sanding Roll" (full title: "4 1/2-Inch x 10yd 80 Grit Adhesive-Backed Sanding Roll"), the BETag surprisingly includes specifications such as "4 1/2-inch" and "10-yard," which were not present in the base tags. The inclusion is reflective of user purchasing behavior, indicating that these specifications are commonly associated with the item. This demonstrates how BETags adapt to user tendencies, though minor discrepancies, such as incorrectly tagging "120 grit" instead of "80 grit," occasionally occur.

Overall, base tags successfully capture core semantic properties based on item information, while BETags provide broader, behavior-driven semantic connections. This broader semantic representation allows BETags to establish more nuanced connections across items, enhancing the relevance and engagement of recommendations in various scenarios.

## D   Implementation Details.

*Dataset Preprocessing.* For the Amazon (Scientific) dataset, we followed the preprocessing steps outlined by Li *et al.* [10]. In the MovieLens-1M dataset, only users with at least 20 interactions and items with at least 30 interactions were retained. Plot summaries for each movie, retrieved from OMDb [1], were used as input for base tag generation in this dataset. For the FreshFood dataset, which contains user browsing data from an e-commerce platform in Taiwan specializing in fresh food, we crawled item details such as titles, descriptions, and keywords from the website. These keywords, referred to as native tags, were used as baseline tags. Few items no longer available on the platform were filtered out during the preprocessing stage.

*BETag Finetuning and Generation.* For the base tag generation, we employed the *gpt-3.5-turbo* model across all datasets. The finetuning of BETag was conducted using the *MaziyarPanahi/Llama-3-8B-Instruct-v0.8* [2] model for the MovieLens-1M and Amazon (Scientific) datasets. For the FreshFood dataset, we utilized the *yentinglin/Llama-3-Taiwan-8B-Instruct* [3] model, which was finetuned on a large corpus of Traditional Mandarin.

We employed LoRA[7] with a rank of 64 for parameter-efficient finetuning. Due to computational constraints, the models were quantized into NF4 format, with finetuning performed in 16-bit floating-point precision on an Nvidia V100 32GB GPU. Each LLM was finetuned individually for the respective datasets, using a learning rate of $10^{-4}$ over 15 epochs.

During finetuning, user behavior sequences were limited to a maximum length of 15 interactions ($n = 15$). Instead of using only the most recent interactions for training, we applied a random cropping strategy, where a window of 15 interactions was randomly selected from each user's full interaction history during each training epoch. This approach differs from the inference stage in downstream tasks, where only the most recent interactions are considered.

*Downstream Tasks.* For the downstream tasks, we performed a grid search using BETag with a range of beam sizes $m \in \{1, \ldots, 40\}$ for multi-beam generation. The optimal beam size was selected based on performance on the validation set, where the second-to-last interaction in each user behavior sequence was used to evaluate model effectiveness.

In item-based retrieval tasks, two different methods were explored. For BM25, we conducted a grid search over the parameters $k_1 \in \{0, 0.5, 1.0, 1.5, 2.0\}$ and $b \in \{0, 0.25, 0.5, 0.75\}$. For BiRank, we set $\alpha = \beta$, and we sampled ten values from the range $[0.1, 0.9]$ for grid search.

For user-based retrieval tasks, we explored different methods of weighting the user interactions, comparing uniform weighting against linear weighting, and selected the best-performing strategy based on validation results.

---

[1]https://www.omdbapi.com
[2]https://huggingface.co/MaziyarPanahi/Llama-3-8B-Instruct-v0.8
[3]https://huggingface.co/yentinglin/Llama-3-Taiwan-8B-Instruct

**Table 6: Qualitative results of BETag**

| Title | Item Information | Native Tags | Base Tags | TagGPT | BETag |
|---|---|---|---|---|---|
| Tender Mercies (1983) | Alchoholic former country singer Mac Sledge makes friends with a young widow and her son. The friendship enables him to find inspiration to resume his career. | Drama, Romance, Western, Music | Country music drama, Redemption story, Alcoholism recovery, Intergenerational friendship, Small-town life, Musical inspiration, Heartwarming film | Redemption story, Alcoholism recovery, Resilience, Country music, Heartfelt performances, Friendship, Musical drama | Coming of age, Quirky characters, Family dynamics, High school drama, Teen romance, Love triangle, Forbidden love |
| The Godfather (1972) | Don Vito Corleone, head of the Corleone mafia, refuses a rival's request to use his influence to sell drugs, sparking a violent conflict between the old ways and modern crime. His son Michael, a WWII hero, initially reluctant to join the family business, is drawn into a mob war that threatens to tear the Corleone family apart. | Action, Crime, Drama, Mafia | Mafia drama, Organized crime, Family loyalty, Power struggle, Generational conflict, New York setting, Betrayal and revenge, Legacy and tradition | Mafia drama, Power struggle, Organized crime, Generational conflict, Family loyalty, Loyalty and betrayal, Crime family, New york city | Mafia drama, Organized crime, Friendship, Adventure, Psychological thriller, Brotherhood, Historical drama, Survival |
| PORTER-CABLE 740000801 4 1/2-Inch x 10yd 80 Grit Adhesive-Backed Sanding Roll | 'brand': 'PORTER-CABLE', 'category': 'Industrial & Scientific Abrasive & Finishing Products Manual Sanding Products Sanding Rolls' | - | Adhesive-backed sanding roll, 80 grit for effective sanding, Industrial-grade sanding product, Portable and convenient roll design, Suitable for manual sanding tasks, Brand: porter-cable, High-quality abrasive material | Adhesive-lined for secure sealing, Long 36-yard roll, Additive manufacturing product, Reliable porter-cable brand, High-quality manual sanding product, Ideal for additive manufacturing, Industrial & Scientific | Adhesive-backed sanding roll, Easy application and removal, Porter-cable brand, Versatile 4 1/2-inch width, Durable 10-yard length, 120 grit sandpaper, Manual sanding products |
| SainSmart PLA-151 PLA Filament (Red) | 'brand': 'SainSmart', 'category': 'Industrial & Scientific Additive Manufacturing Products 3D Printing Supplies 3D Printing Filament' | - | High-quality PLA filament, Sainsmart brand, Red color option, Industrial grade, Ideal for 3D printing, Reliable printing supplies, Additive manufacturing material | High-quality PLA filament, Industrial grade, Compatible with 3D printers, Vivaplex brand, Ideal for industrial 3D printing, Reliable printing results, Industrial strength adhesive | High precision printing, Consistent dimensional accuracy, High-quality PLA filament, Vibrant yellow color, Compatible with various 3D printers, 1.75 mm diameter |

