# OpenReview forum: "BETag: Behavior-enhanced Item Tagging with Finetuned Large Language Models"
_ACM.org/TheWebConf/2025/Conference — WWW 2025 Poster_

### Official Review · Reviewer_KE8Q · 2024-11-21

**Novelty:** 4
**Technical Quality:** 4

**Review:**

Pros

1. This paper addresses a practical problem by generating tags for items using LLM, considering both item semantics and user preference. A behavior-enhanced fine-tuning is proposed to consider the user preference for a more accurate tagging process.

2.  The paper is generally well-written and easy to follow.

3. Extensive experiments on three public datasets demonstrate the effectiveness of generated tags on four downstream tasks.

Cons/Questions

1.  Are the tags treated as independent tokens or composed of individual word tokens?

2. The effectiveness of the BETag is verified through its integration into four downstream tasks.  I think it is also necessary to evaluate the quality of the generated tags, e.g. its consistency with the base tag or title, or some language quality metrics like Bleu, etc. Besides, for the recommendation task, there is only one baseline (UniSRec) used. i suggest the author add one or two more baselines.

3. Can case studies be provided to illustrate the difference between the base tags and the behavior-enhanced tags?

4. The implementation code is not available.

**Questions:**

Refer to the cons above for questions.

**Reviewer Confidence:**

3: The reviewer is confident but not certain that the evaluation is correct

**Scope:**

4: The work is relevant to the Web and to the track, and is of broad interest to the community

---

### Official Review · Reviewer_g2xL · 2024-11-26

**Novelty:** 5
**Technical Quality:** 4

**Review:**

The paper introduces a novel framework for automatic product tagging in e-commerce. The proposed method generates "Behavior-enhanced Tags" (BETags) by refining initial tags created by LLMs with user behavior data. This approach improves tag relevance and alignment with user preferences, enhancing search optimization, recommendation systems, and product discoverability. BETag is evaluated on three datasets (Amazon, MovieLens-1M, and FreshFood), showing superior performance over human-annotated tags and prior automated methods, while maintaining scalability and efficiency.

Pros:

- Innovative Integration: Combines LLMs with user behavioral data, enhancing the contextual relevance and personalization of item tags.
- Strong Experimental Validation: Demonstrates effectiveness across multiple datasets and evaluation metrics, outperforming state-of-the-art methods.
- Clear Writing: Well-structured and easy to follow.

Cons:

- Some related work was not compared, such as [1].
- Item Tagging Sparsity: The issue of item tagging sparsity remains, with performance slightly trailing behind TagGPT.

[1] Genki Kusano et al. GA-Tag: Data Enrichment with an Automatic Tagging System Utilizing Large Language Models. ICDE'24

**Questions:**

- Why are there five different finetuning answers in Figures 2 and 3? Where are the four labels other than the next item from?
- How can we ensure that, in Phase 3, the next most likely item a user interacts with—given only one item in the history—is similar to the history item, rather than resulting in a prediction that reflects a broader shift in interests by the LLM?
- The performance compare to more baseline models.

**Reviewer Confidence:**

2: The reviewer is willing to defend the evaluation, but it is likely that the reviewer did not understand parts of the paper

**Scope:**

4: The work is relevant to the Web and to the track, and is of broad interest to the community

---

### Official Review · Reviewer_w9zn · 2024-12-02

**Novelty:** 4
**Technical Quality:** 3

**Review:**

The paper proposed a novel approach for automatic product tagging in e-commerce using fine-tuned large language models (LLMs). This approach generates behavior-enhanced tags (BETags) by refining base tags derived from LLMs with user behavior data.

[Pros]

[P1] Novelty: The paper incorporates user behavior data into the tagging process, ensuring tags align better with actual user preferences and interests.

[P2] Versatility and significant performance: Experiments on various datasets demonstrate that BETags outperform both human-annotated tags and the baseline method. It also shows consistent performance in both recommendation and retrieval tasks.

[P3] Effective component: The effectiveness of behavior-enhanced fine-tuning is proved through an ablation study.

[Cons]

[C1] Limited baseline comparisons: The paper does not provide comparisons with state-of-the-art retrieval and recommendation models that do not rely on tagging. Including such baselines would strengthen the claim of BETag’s effectiveness in broader scenarios.

[C2] Robustness to Data Sparsity and Noise: The authors do not explicitly address the challenges posed by sparse or noisy user behavior data. It is unclear how the model performs when the behavioral data is incomplete or contains inconsistencies, which are common in real-world applications.

[C3] Reproducibility: The authors have not provided the implementation or code necessary to reproduce the results, which raises concerns about the reproducibility and independent validation of the findings.

**Questions:**

Based on my cons, I have a few questions.

[Q1]. To address [C1], could the authors provide comparisons with state-of-the-art retrieval and recommendation models that do not rely on tagging?

[Q2]. To address [C2], how does BETag perform in scenarios with sparse or noisy user behavior data? Are there specific components in the methodology designed to address these challenges, and if not, how might future work incorporate such robustness?

[Q3] To address [C3], could the authors share the implementation code or more detailed steps for reproducibility? The lack of reproducibility is the reason for my low technical quality score for this work.

**Reviewer Confidence:**

3: The reviewer is confident but not certain that the evaluation is correct

**Scope:**

4: The work is relevant to the Web and to the track, and is of broad interest to the community

---

### Official Review · Reviewer_4xw7 · 2024-12-02

**Novelty:** 7
**Technical Quality:** 6

**Review:**

The paper shows the limitation of current item tagging strategies and demonstrates a new approach to generating item tags that are enhanced by using user interaction behavior with items.

They use a 3 step approach, which first starts with using a vanilla LLM for generating tags from an item’s metadata, such as title and description. Second, they fine-tune an LLM using sequences of user item interactions, where the items are replaced with the tags generated in the first stage. Third, they use the fine-tuned LLM to generate tags for the item catalog. Tags generated using this approach contain user specific preferences. In the third stage they also use multi-beam search to generate a more diverse set of tags. This helps address sparsity and generates a more diverse set of tags for the items in the catalog.

The authors validate this technique on 3 different datasets and compare the efficacy of this technique for multiple downstream applications with other tag generation techniques. For evaluating the efficacy on recommendation tasks, they compare this technique with SASRec/UniSRec. The authors have also run ablation studies that demonstrate the benefit of combining fine-tuning with multi-beam search.

Strong Points
1. The paper is well written and easy to follow
2. The use of user information to augment statically generated tags for items is a novel and seemingly useful idea
3. Clear evidence across three datasets that shows the performance improvement, especially against other tagging strategies
4. The ablation studies demonstrated very well the benefits of fine-tuning using user behavior and multi-beam generation

Weak points
1. For the recommendation task, it might have been useful to compare with additional baselines using newer models such as Recformer[10] and CALRec (https://arxiv.org/pdf/2405.02429). Alternatively, some discussion on why only SASRec was used as a baseline as opposed to Recformer and CALRec would be helpful.
2. Some more details on how randomization was used for multi-beam search would have been useful
3. Some discussion on real world implications, such as cost / effectiveness for large catalog sizes and the cold start problem, would have been very informative.

**Questions:**

Addressing the weak points listed in the previous section would be very helpful.

**Reviewer Confidence:**

3: The reviewer is confident but not certain that the evaluation is correct

**Scope:**

4: The work is relevant to the Web and to the track, and is of broad interest to the community

---

### Official Review · Reviewer_vGRD · 2024-12-03

**Novelty:** 4
**Technical Quality:** 3

**Review:**

### Summary

The paper proposes a product tagging framework designed to generate contextually relevant tags that better align with real-world user preferences. The framework begins by generating initial tags for each product, converting the items in a user's history into corresponding tags. These tags are then used to construct a user behavior sequence, which is leveraged to fine-tune a large language model (LLM). This fine-tuning enables the LLM to predict the expected next user behavior in terms of tags. Using this fine-tuned model, the framework generates enhanced, context-aware tags for each product.

### Strength

The proposed framework demonstrates effectiveness compared to baseline models.

### Weakness

- The framework employs an LLM but does not compare its performance against other LLM-based recommendation frameworks (e.g., P5, TIGER, IDGenRec, etc.), limiting its competitiveness. Instead, it only compares with SASRec and UniSRec, where the observed performance improvements are not particularly convincing.

- It is unclear whether the model, trained on sequences of item tags (with input prompts containing multiple items), is effectively utilized to generate improved tags for individual items (with input prompts containing a single item). This raises concerns about the generalizability of the approach for individual item tagging. This concern is showing on Table 4 where multi-beam actually performs worse on FreshFood dataset.

**Questions:**

None

**Reviewer Confidence:**

3: The reviewer is confident but not certain that the evaluation is correct

**Scope:**

3: The work is somewhat relevant to the Web and to the track, and is of narrow interest to a sub-community